# Perspectives of Adults with Intellectual Disabilities on Quality of Life: A Qualitative Study

**DOI:** 10.3390/ijerph21091186

**Published:** 2024-09-06

**Authors:** Pavlos Kapsalakis, Evdoxia Nteropoulou-Nterou

**Affiliations:** 1Primary Special Education School, Ministry of Education, Religious Affairs, and Sports, 15122 Marousi, Greece; 2Department of Early Childhood Education, School of Education, National and Kapodistrian University of Athens, 10676 Athens, Greece; ederou@ecd.uoa.gr

**Keywords:** intellectual disability, quality of life, occupational quality of life, o-qol, occupational therapy, occupational justice, social inclusion

## Abstract

Experiences of occupational participation of adults with Intellectual Disabilities (IDs) were explored through the lens of the Model of Occupational Justice (MOJ) and Critical Theory in order to shape and develop an occupation-centered model of quality of life (QoL). This qualitative study involved thirteen adults with IDs (*N* = 13). A semi-structured interview, constructed based on MOJ and Critical Theory principles, was administered to explore perspectives on QoL, as well as injustices regarding occupational participation. The interviews were analyzed using QSR NVivo8 and followed a content analysis methodology. A preliminary model of Occupational Quality of Life (O-QoL), with an everyday occupations core component, has been formed. The model includes three core O-QoL domains: (i) social well-being, (ii) emotional–physical well-being, and (iii) material adequacy. Key indicators of O-QoL were identified as leisure and social activities, while socioenvironmental factors such as occupational deprivation were noted as aggravating. Specific occupations, including leisure activities, physical exercise/sports, art, video games, and vocational training, were found to be beneficial for O-QoL. Moreover, the importance of promoting and supporting the rights of people with IDs for employment, independent living, and sexual expression was highlighted. The model of O-QoL (version 1) could be a valuable alternative conceptual framework of QoL in the field of IDs; however, further research is needed to validate and refine the model.

## 1. Introduction

Life experiences of people with Intellectual Disabilities (IDs) are often characterized by social marginalization, educational and professional exclusion [1], and pervasive stereotypes about IDs [1,2]. Most individuals with IDs live with their families, participate in fewer social activities with friends, and have far fewer mutual friendships compared to their peers [3]. Additionally, they spend more time alone [4], are less socially active, and their friends are usually other people with IDs [5]. Individuals with IDs were often subjected to dehumanizing treatment under the medical model [6], which focuses on deficits and pathology. The emergence of the social model of disability [7] challenged these notions by highlighting societal barriers rather than individual impairments, advocating for inclusion and empowerment. This shift reflects an ongoing struggle to address systemic inequities and promote full participation for people with IDs, emphasizing their rights and contributions to society.

Quality of life (QoL) is a multidimensional construct that reflects an individual’s overall well-being. The concept of QoL has evolved significantly over time, especially in the context of people with IDs. Initially, QoL for individuals with IDs was guided by the medical model [7], emphasizing diagnosis, treatment, and management of symptoms and behaviors. This approach aimed to evaluate the extent to which medical interventions could alleviate symptoms or improve functional abilities. This emphasis on “*normalization*” [8,9] often led to marginalizing people with IDs, as their experiences were framed primarily in terms of their impairments. By the 1980s, the concept of QoL began to expand to include broader dimensions such as social relationships, personal development, and emotional well-being. The paradigm shift in the 1990s was pivotal, with the introduction of holistic and person-centered approaches [10]. Schalock et al. played a key role in this evolution by proposing a multidimensional model that addresses QoL specifically for people with IDs and includes eight core domains [11]: emotional well-being, interpersonal relations, material well-being, personal development, physical well-being, self-determination, social inclusion, and rights. This model underscores the importance of individual experiences and subjective perceptions, marking a significant departure from earlier, medical interpretations of QoL, provides a comprehensive understanding of what constitutes a fulfilling life, and guides efforts to enhance QoL and promote social inclusion for people with IDs.

Occupational therapy (OT) helps people participate in *occupations*, i.e., activities of daily living that are important and meaningful to them, which can improve their overall well-being [12]. Occupational therapists support people with IDs in various settings (e.g., home [13], school [14], vocational training, and employment programs [15]), while OT interventions also address socioenvironmental conditions [16], such as limited access to education and employment, restricted social life and social marginalization, inadequate social support networks, poor housing conditions, and transportation barriers, that affect participation and engagement in a broad range of activities and impact health and QoL [17]. Scientific evidence indicates that activities of daily living particularly influence people’s health and well-being. Specifically, the relationship among desirable/undesirable roles in occupations [18], social participation [19], and kinds of exclusion due to external—out of the individual’s control—factors, such as occupational deprivation [20], impacting physical health and QoL has been highlighted.

OT models are important theoretical concepts shaping clinical reasoning and connecting theory with clinical practice [21]. The theoretical foundation of the *Occupational Justice* Model (MOJ) suggests eliminating social exclusions and promoting well-being via active participation in meaningful occupations [22,23] since people want and ought to engage in pleasing and essential occupations of their choice for better health and QoL [24]. Given that the concept of QoL has evolved into a social construct defining sociopolitical practices so that citizens with IDs have the right to access/participate equally in activities of everyday life and enjoy the same human rights and level of QoL as any other member of the society [25]; the core principles of the MOJ align well with the contemporary approach of QoL, and OT can serve as a sociopolitical practice [26] addressing *occupational injustices* [22], such as (i) *deprivation* [20] (i.e., denied participation in meaningful activities), (ii) *alienation* (i.e., lack of personally meaningful experiences), (iii) *marginalization* (i.e., restricted choices in roles and cultural identity), and (iv) *imbalance* (i.e., unequal opportunities for activities). At the same time, OT can foster occupational rights [27]: (i) right to social inclusion through occupational participation, (ii) right to meaningful occupational experiences, (iii) right to take desired roles in occupations and maintain personal and cultural autonomy through free choices for occupations, and (iv) equal rights for a variety of ways of participating in occupations, thereby promoting and enhancing QoL.

By applying the MOJ’s principles, our study suggests an alternative, occupation-centered, and more inclusive understanding of QoL for people with IDs. From the point of view of individuals with IDs, occupation-centered QoL has never been explored yet, and regarding the complexity of the QoL concept, there is an imperative need to better understand perceived occupation-centered QoL for this population. To address this gap, the current study set out to use qualitative methodology to gain knowledge about the occupational participation experiences and QoL perceptions of adults with IDs. Therefore, the first objective of this study was to explore their occupations and highlight factors that affect, inhibit, and/or limit access and active participation. The second objective was to explore their perceptions about the concept and meaning of QoL. And finally, the third objective was to identify factors that affect QoL, explore the relationship between occupational participation and QoL, and elicit their perspectives about its improvement.

Conclusively, the main purpose of this study was to produce foundational knowledge and construct a preliminary theoretical framework of QoL in the field of IDs based on everyday occupations. This preliminary conceptual model of perceived QoL focused on the occupational needs of individuals with IDs, as well as the factors influencing their occupational participation and the impact of occupations on their QoL, would significantly contribute to a more comprehensive and nuanced understanding of the critical role that occupations play in the daily lives of this population. Moreover, our suggested model could enhance and extend the existing OT literature on QoL, offering valuable insights into the intersection of occupations and QoL for people with IDs. Although substantial international research has explored QoL among individuals with IDs (e.g., Schalock’s foundational research is a significant contribution to the field), much of this work has not focused specifically on the occupational aspects of their daily lives. Moreover, to our knowledge, no similar research has been conducted in Greece, and existing studies in the field of OT and QoL have predominantly focused on other populations, such as cancer patients [28] and older adults [29]. Therefore, the conceptualization of an occupation-centered model of QoL based on the perceptions of individuals with IDs would offer an alternative theoretical framework for understanding and enhancing QoL. This innovative model of Occupational Quality of Life (O-QoL) could lead to the development of more occupation-centered practices promoting social inclusion, while within the fields of Occupational Science and OT, it has the potential to advance clinical reasoning, evaluation, intervention, and research, thereby providing a more inclusive alternative to foster equal participation for individuals with IDs.

## 2. Materials and Methods

### 2.1. Study Design and Ethics

Qualitative studies excel at offering a deeper insight into how people’s views may vary, the attitudes and beliefs they hold, and the factors that shape specific perspectives [30]. The semi-structured interview was used as a research tool in this study. This type of interview is a qualitative research method that has a predefined set of open questions, which can be modified depending on the perception of the interviewer in order to prompt discussion. The wording of the questions can be modified, clarified, and omitted, and also additional questions can be asked according to each respondent, giving an opportunity for the interviewer to explore particular themes or responses further [31]. Therefore, a qualitative descriptive study was conducted using semi-structured interviews to explore experiences of occupational participation and perspectives on QoL. Semi-structured individual interviews were chosen as the mode of data collection over alternatives (e.g., questionnaires), as they encourage discussion and provide extensive insight into participants’ beliefs and experiences.

The formation of the Interview Guide was influenced by Critical Theory [32] and the MOJ [22,23], both of which aim to critique and challenge the power structures and ideologies that perpetuate inequality and oppression in society. Critical Theory emphasizes the need for societal change through the examination of culture, politics, and economics, while striving to uncover the ways in which dominant ideologies shape and control the experiences of marginalized groups, such as individuals with IDs [33]. Similarly, the MOJ addresses injustices and inequalities in people’s occupations by applying these critical perspectives to promote occupational equity. Initially, a pilot version of a semi-structured Interview Guide was developed and administered on a trial basis to a female participant. The Interview Guide was designed with a dual focus to comprehensively address key aspects of the participants’ daily experiences. First, it sought to explore the full range of activities that participants engaged in within a Vocational Education Foundation (VEF), aiming to capture the breadth and depth of their involvement in this structured environment. Second, the guide aimed to gain a deeper understanding of how participants spent their time beyond the VEF, with particular emphasis on their routines and interactions in home and leisure settings. This approach was intended to provide a holistic view of their daily lives, encompassing various environmental contexts in which they operate. By adopting this approach, we aimed to gather optimal, thorough, and holistic insights into the daily lives and experiences of the participants.

In response to the initial outcomes and reflections on our data collection process, we developed an adapted and revised version of our original Interview Guide. The primary objective of this revision was to enhance the quality of the interviews by refining the phrasing of questions. Specifically, we aimed to reduce the extent to which our questions might inadvertently guide respondents, thereby allowing for more authentic and uninfluenced responses (e.g., “*do you like coming here to school?*” was rephrased as “*how do you feel about coming here to school?*”, etc.). Simplifying the language used in the questions was also a key priority, ensuring that the content was easily comprehensible for all participants, particularly given the diverse cognitive abilities of those involved (e.g., the term “*school*” was used by the facilitator to refer to the VEF because the participants where more familiar with that terminology, etc.). To address the abstract nature of terms such as “*ποιότητα ζωής*” (quality of life) and “*ευζωία/ευ ζην*” (well-being), we undertook a comprehensive review of Greek terminology as used by participants. We identified and employed alternative expressions, such as “καλή ζωή” (good life) and “*χαρούμενη ζωή*” (happy life), which were found to be conceptually appropriate and meaningful within the cultural context. This approach ensured that the terms used in the interviews were both comprehensible and reflective of participants’ lived experiences. The selection and interchangeability of these terms were carefully considered to maintain conceptual accuracy and relevance throughout this study. Moreover, we restructured the interview format to allow for a more organic and less rigid progression of topics. This approach encouraged a more conversational style, enabling participants to express their thoughts more freely and naturally. Consequently, the interviewer was required to demonstrate increased flexibility and responsiveness, adapting to the flow of the conversation as it evolved. This methodological shift was particularly important in ensuring that the interviews were participant-centered, rather than strictly adhering to a pre-set sequence of questions. For example, the conversation might begin with a topic of the participant’s interest, such as a favorite TV series, and then naturally shift from discussions about the VEF’s activities to leisure activities, or vice versa, based on the flow of the conversation.

The interviews were originally conducted in Greek, and a rigorous translation process was implemented to convert the content into English. This process involved multiple stages of translation and back-translation [34] to ensure fidelity and accuracy. The primary researcher, in collaboration with a bilingual expert familiar with both Greek and English and experienced in disability studies, undertook the translation. The translation focused on selecting terms that were culturally and conceptually equivalent, aiming to preserve the nuance of the original Greek terms. The final choice of terms was guided by a consensus between the researcher and the supervising professor, with careful consideration given to how these terms would be understood in the context of participants’ lived experiences.

Informing and obtaining consent from people with IDs can be challenging, as there is no protocol for obtaining informed consent, either conceptually or as a procedure, from this particular population [35]. In order for the consent to be valid, the prospective participant is required to assess the current situation, have sufficient information, understand the information given, be able to weigh the advantages and disadvantages, and communicate a voluntary decision free from any coercion [36]. Thus, obtaining consent from participants with IDs presents serious ethical challenges for the researchers. Disorders in attention, memory, and/or the ability to recall information material may create confusion about the purpose of the research and the consequences of participating in it, and at the same time, the researcher should avoid coercion while explaining them [37]. At first, after being verbally briefed on the study, participants and their parents were also provided with written informed consent. Afterward, participants and their parents who expressed voluntary interest for this research were requested to give verbal and written consent, respectively. Specifically, parents agreed for their children’s participation by signing a form that confirmed voluntary participation, confidentiality, and data protection. Finally, participants who agreed to participate were informed that they could end the interview at any stage, withdraw their consent, and leave whenever they felt they wanted to, and also that they had no obligation to complete the procedure or give any explanation for their withdrawal. To ensure the validity of the informed consent process, we established a supportive environment where participants and their guardians could freely ask questions and raise concerns. This approach included providing ample time for discussion to address any uncertainties. Additionally, we implemented a continuous and robust monitoring system throughout the interview process to detect and address potential misunderstandings or ethical issues. The researcher remained vigilant in identifying signs of confusion or distress and was prepared to offer immediate clarification and support. Finally, the responses obtained from participants were carefully evaluated and confirmed to be valid, reflecting a clear understanding of the questions posed and the information provided. This process contributed to the overall validity of the participants’ consent.

### 2.2. Participant Recruitment

The interviews took place in the Attica region of Greece at a VEF for adults with IDs. VEFs are specialized institutions designed to provide comprehensive support and services to adults with IDs. VEFs offer a structured environment where individuals can engage in vocational training and therapeutic programs aimed at enhancing their skills, independence, and overall QoL. VEFs operate as multi-functional daycare centers that integrate vocational training, therapeutic care, leisure activities, etc. These institutions cater to adults with IDs who are typically beyond the age of compulsory education. The primary objective of VEFs is to facilitate the acquisition of practical skills that can lead to employment, either within the foundation or in the broader community. To achieve this, VEFs offer various vocational programs that are tailored to the abilities and interests of each participant. These programs may include training in areas such as crafts, food preparation, gardening, or other hands-on activities that have the potential to translate into employable skills. In addition to vocational training, VEFs provide a wide range of services designed to support the physical, social, and psychological well-being of participants. These services often include leisure activities, occupational therapy, physical rehabilitation, social skills development, psychological counseling, and other forms of therapy that aim to enhance the participants’ overall functioning and adaptability. The leisure and therapeutic programs are integrated into the daily schedule and are personalized to meet the specific needs of each individual, ensuring that they receive holistic care. Participants in a VEF typically attend the facility on a daily basis, where they engage in a structured routine that balances work-related tasks with therapeutic activities. The environment is designed to be supportive and nurturing, fostering a sense of community and belonging among participants. Staff at VEFs are trained professionals who provide continuous supervision, guidance, and support, ensuring that each individual can progress at their own pace. The impact of VEFs extends beyond vocational training. By providing a safe and supportive environment for learning and development, these institutions can play a crucial role in promoting the social inclusion and independence of individuals with IDs. The skills acquired through VEFs’ programs can lead to meaningful employment opportunities, which may, in turn, enhance participants’ self-esteem and QoL.

Male and female participants were recruited via *convenience sampling* [38] to purposefully take part in research about their perceptions of QoL. Facilitator contacts and face-to-face invitations in the VEF were used to recruit participants. The participant recruitment procedure mainly focused on the verbal communication ability of individuals with IDs, as well as the expressed desire for participation in this study. Specifically, inclusion criteria were (i) diagnosis of IDs or other medical conditions with ID comorbidity; (ii) attending the VEF; (iii) ability of verbal communication; and (iv) having attained eighteen years of age. In total, 13 participants (n = 9 males; n = 4 females) were recruited.

### 2.3. Data Collection

Data collection occurred at conveniently selected locations within the VEF. Individual semi-structured interviews were conducted, employing open-ended questions designed by the facilitator to foster in-depth discussion. The Interview Guide, detailed in Table 1, was developed based on the existing literature related to the MOJ, Critical Theory, and QoL, with a focus on participants’ everyday activities. The guide comprised three main categories of questions aimed at eliciting participants’ experiences of occupational participation and their perceptions of well-being and QoL. The key categories of questions included the following:*Occupational Experiences and Perceptions of Occupational Participation*: This category sought to explore participants’ engagement in various activities, their roles within these activities, and their emotional responses to participation or lack thereof. Examples of questions in this area include “In what kind of activities do you participate?”, “How do you feel when you participate or do not participate in activities?”, etc.*Perspectives and Perceptions of QoL*: This category aimed to understand participants’ definitions of QoL, their methods for assessing it, and the factors influencing it. The questions in this area such as “What does QoL mean to you?”, “Which factors affect your QoL?”, etc., were used to gather insights.*Perceptions About Ways to Promote or Improve QoL*: This category focused on identifying potential strategies for enhancing QoL from the participants’ perspectives. This area of questions included “How can your QoL be improved?”, “How could you promote your QoL?”, etc.

The facilitator encouraged participants to express their views freely and probed further when elaboration was insufficient. If the conversation deviated from the guide, the facilitator evaluated its relevance and redirected the discussion as needed to maintain alignment with the Interview Guide’s objectives. Each category was designed to capture detailed insights into the participants’ experiences and perspectives, ensuring a comprehensive understanding of their QoL and occupational engagement.

Thirteen individual semi-structured interviews were conducted, each lasting an average of fifty-three minutes. All interviews (*N* = 13) took place over a three-day period in mid-December 2020 and were audio recorded using a digital recorder. Participants were pseudonymized and assigned unique interview identification codes. The audio recording began after obtaining verbal consent from each participant. Prior to starting the interview, the facilitator provided a brief introduction, reminding participants that their audio was being recorded and assuring them of the confidentiality of their responses. After completing the interviews, participants were debriefed on the study’s purpose, and any major misconceptions or inaccuracies about the nature/purpose of the interviews were corrected. Demographic information was collected through questionnaires administered to parents.

### 2.4. Data Analysis

In this qualitative study on the QoL of adults with IDs, careful attention was given to the processes of data collection and analysis to enhance the trustworthiness of the findings. The primary researcher conducted all interviews using a voice recorder for audio documentation and supplemented this with written notes to capture additional observations and contextual details. Additionally, a Participant Demographic Form was utilized to gather comprehensive information on participants’ demographic characteristics. All recordings were transcribed verbatim and proofread by the primary researcher. The transcripts were then imported into QSR NVivo8 for processing. To achieve familiarity with the data, transcripts were read meticulously multiple times. The theme development process was deeply rooted in a systematic content analysis approach, which facilitated the identification and organization of key themes within the qualitative data. The content analysis method [39,40] was particularly well-suited to this research as it allowed for a nuanced exploration of the participants’ daily lives and their perceptions of QoL. The process unfolded through several critical stages, each contributing to the emergence of meaningful themes that reflect the lived experiences of adults with IDs. We began with an open coding process, where the entire data set was meticulously coded without reliance on a pre-existing coding frame. This inductive approach allowed the codes to emerge organically from the data, ensuring that the analysis was grounded in the participants’ own narratives and experiences. The initial focus was on identifying broad categories of daily occupations and activities, such as leisure, education, and vocational training, which served as the primary organizational structure for the coding process. As the coding process advanced, distinct categories were systematically developed to organize the participants’ references to QoL. 

Critical Theory and the MOJ were employed as interpretative lenses during the coding process. These frameworks guided the analysis by providing a conceptual foundation for understanding the participants’ experiences in relation to broader sociopolitical structures and issues of justice and equity. While the primary focus of the coding was on the participants’ occupations, these theoretical perspectives enriched the analysis by highlighting the underlying power dynamics and systemic issues that influenced the participants’ QoL. Additionally, the MOJ framework played a crucial role in identifying and categorizing instances of occupational injustice within the data. Some of the framework’s concepts, such as occupational deprivation, alienation, marginalization, imbalance, etc., were used as additional coding categories. This allowed for the analysis to connect participants’ experiences of occupation with broader issues of social justice, illustrating how systemic barriers and inequities impacted their daily lives and QoL.

Following the initial coding and categorization, the primary researcher engaged in a process of theme refinement. This involved reviewing and re-coding the data to ensure that the themes fairly reflected the participants’ experiences and were coherent across the data set. The collaboration with the supervising professor was instrumental during this stage, as it provided a critical review and helped resolve any discrepancies in the coding. The final themes were developed through an iterative process of synthesis, where the researcher integrated the coded data into broader thematic narratives. These themes not only captured the participants’ experiences and perceptions of QoL but also illuminated the ways in which their daily occupations and encounters with occupational injustice shaped their lives. The use of both inductive coding and theoretical frameworks ensured that the themes were both data-driven and conceptually robust, providing a comprehensive understanding of the QoL of adults with IDs. This detailed and rigorous thematic development process was central to this study, ensuring that the analysis was grounded in the participants’ voices while being informed by relevant theoretical perspectives. The resulting themes offer valuable insights into the complex interplay between occupation, QoL, and Occupational Justice for adults with IDs.

To support the credibility of the analysis, two rounds of coding were performed by the primary researcher, which enabled the assessment and improvement of *intracoder reliability* [40]. Intracoder reliability refers to the consistency with which the same researcher codes the data at different points in time. It measures whether the researcher would apply the same codes to the data if they were to revisit it after a certain period. Assessing intracoder reliability can be valuable in promoting reflexivity, as it encourages researchers to reflect on their coding decisions and potential biases, while serving as a useful tool for ensuring that the coding process remains stable and consistent over time. This step involved revisiting the data to ensure consistency in the interpretation of the participants’ responses. Throughout this process, the primary researcher worked closely with the supervising professor, who provided ongoing guidance and critical oversight. Regular discussions were held to reflect on the coding process, addressing any discrepancies, and ensuring that the emerging themes reasonably represented the participants’ lived experiences. This collaboration was crucial in refining the analysis and helped to balance the potential biases that might arise from having a single researcher involved in both data collection and analysis. To further ensure this study’s trustworthiness, detailed documentation of the research process was maintained, which included notes on coding decisions, reflective thoughts, and the rationale behind the development of themes. Although *member checking* [41] was only informally incorporated, with the primary researcher summarizing and confirming participants’ interpretations during data collection, the rigorous approach to reflective thinking and collaboration with the supervising professor, who served as a *critical friend* [42], were crucial in enhancing the credibility of the findings. A critical friend refers to an external individual who provides constructive feedback and rigorous scrutiny throughout the research process. This person, who is not directly involved in this study, offers an objective perspective on the research design, methodology, and interpretations. The role of the critical friend is to challenge assumptions, identify potential biases, and ensure the research maintains methodological rigor, thereby enhancing the overall trustworthiness and credibility of the findings. Conclusively, the final themes were meticulously compared with the existing literature to validate their relevance and ensure alignment with established knowledge in the field.

## 3. Results

### 3.1. Participation and Sample Characteristics

Thirteen adults with IDs (*N* = 13) participated voluntarily in this study. As shown in Table 2, the sample consisted of nine men and four women (aged 18–37 years), the vast majority of which had graduated from Special Education Schools (*n* = 11). Most participants attended the VEF’s carpentry (*n* = 4) and bookbinding (*n* = 4) workshops. Furthermore, participants’ total training years in the VEF ranged from one to twenty-three years, while seven people trained there for more than ten years. Finally, regarding participants’ living conditions, most of them lived with their parents (*n* = 10) and had no personal space (*n* = 9) as they either shared their room or did not have their own room at all.

### 3.2. Defining Quality of Life: Perspectives and Perceptions of People with Intellectual Disabilities on the Concept of Quality of Life

The content analysis was meticulously applied to various segments of the data, resulting in the identification of four primary categories: (i) *Meaning and Definitions of QoL* captured the participants’ personal interpretations and the values they associated with a good life; (ii) *Beneficial and Aggravating Factors* encompassed the elements that participants identified as either enhancing or diminishing their QoL, providing valuable insights into the conditions and experiences that influenced their overall well-being; (iii) *QoL Assessment Views and Criteria* involved the participants’ perspectives on the appropriate metrics and criteria for assessing QoL, highlighting the measures they deemed most relevant to their lived experiences; and (iv) *QoL Enhancement Views and Criteria* focused on the participants’ suggestions for improving their QoL, revealing their priorities and the specific changes they believed would lead to a more fulfilling life.

Based on the participants’ views in this preliminary study, we developed our proposed model of *Occupational Quality of Life* (O-QoL) (version 1). This model suggests that the core component of QoL is *occupation* (i.e., activities through which personal/social development and improvement of an individual’s personality is achieved) and includes three domains: *social well-being*; *emotional and physical well-being;* and *material adequacy*; these interact with the environment/context (Figure 1).

Therefore, relying on the perspectives of adults with IDs, the model of O-QoL (version 1) defines *quality of life* as follows:


*The individual’s general feeling of well-being, which is determined by the involvement and active participation in desirable and essential for the individual occupations, i.e., every day activities that promote the development of one’s personality on a personal, social, emotional, physical and material level.*


#### 3.2.1. An Occupation-Centered Model of Quality of Life

Generally, participants’ perceptions in defining the term *quality of life* formed four main themes (Table 3):

*Personal/social development and improvement through activities*, the *core component of O-QoL*, is centered on how participation in activities contributes to personal and social development. It underscores that meaningful engagement in these activities is essential for achieving individual goals and fostering social interactions. This concept emphasizes the importance of equitable access to diverse opportunities for personal advancement and social involvement, integrating individual achievements with social enrichment:

*“S3: …I really enjoy making stuff and doing crafts. We also play games like tennis and bowling on the Wii; bowling is definitely my favorite. We have a football team too! In every game I score three goals because I’m really into football. I play as an attacker and always look forward to game days so I can score a bunch of goals!”* (I3).


*“R: …What does it mean to you to live well, to have a good life?*


*S6: We have to move forward in life, make it better and reach high. We need to have many friends, and have fun, but everyone should have a good time.”* (I6).

Overall, QoL is perceived by the respondents as a complex concept, and is mainly approached through leisure activities:


*“R: What does a good life mean to you?*



*S9: Difficult to answer.*



*R: How is your life?*



*S9: Good, good, very good.*



*R: What makes a good life?*


*S9: Going for walks and trips, going for a coffee at the neighborhood square, going to the sea in the summer.”* (I9).

Other participants also highlight the importance of academic advancement and employment, along with their leisure desires, reflecting a broader understanding of QoL that extends beyond immediate leisure activities and contributes to personal growth and development:

*“S2: …to meet new people, travel, do more sports and study… I want to be able to go to a proper school and do what I really like, cooking and pastry.”* (I2).

*“S11: I wish I could go out more often, like taking the car to visit friends and relatives, either at their homes or for coffee. I also enjoy going to the Mall. I want to become a barista and work at the Mall. I had a friend who worked there, and he told me it was great, but he has since moved to the island.”* (I11).

The individuals’ sense of well-being is reflected in their daily engagement in fulfilling activities such as vocational training, enjoying social time with friends, etc. These activities support personal development and emotional satisfaction, and meet physical and material needs, thereby enhancing their overall QoL on multiple levels:


*“R: How is your life going?*



*S5: It’s great, I’m doing very well.*



*R: What makes you feel this way?*


*S5: I have my food, I have my friends, we have a good time. I also like coming here to school and working in the Bookbinding workshop. I can’t think of anything else.”* (I5).

*Social well-being*, the *First Domain of O-QoL*, pertains to the qualitative aspects and dynamics of social relationships, including interactions with friends, family, and community members. It emphasizes the significance of sustaining and nurturing these relationships and highlights the value of opportunities for social engagement and community involvement. This concept underscores how maintaining robust social connections and actively participating in social and communal activities contribute to an individual’s overall sense of well-being:

*“S10: …there’s no one in my neighborhood to hang out with. I do have friends at school; they’re nice kids, but we don’t meet outside of school because we live far apart. I’d like to hang out with them outside of school, but it’s difficult. It would be nice to go for walks, run, or exercise together.”* (I10)

Social well-being was associated with individuals’ interpersonal relationships with family, friends, and the community. Social relationships and social interactions with friends [43]:
*“S4: …I’d rather be at school; I don’t do anything at home. My friends are here, we hang out and talk a bit.”* (I4);
and family members [44]:
*“S7: I wish I could go out for coffee with my dad and my older sister. But now she has a little girl and doesn’t get out of the house much. She had a baby, so I’m an uncle now (super excited)! I do go see her at home, though. This Wednesday, when I get some time out, I’m planning to visit her and have coffee at her place.”* (I7);
as well as social contribution [45] and community service [46]:
*“S3: …I like it here. I love working, playing, and helping people.” (I3)*
*“R: …What does having a good life mean to you?*
*S11: It means loving each other, taking care of one another, and having peace.”* (I11);
were identified as factors composing the social well-being domain of O-QoL.

Emotional well-being refers to an individual’s ability to manage emotions, maintain a positive self-concept, and experience psychological stability and satisfaction through activities:


*“R: What do you think is very important for having a good time?*


*S3: To be happy, to laugh, to play with my brother or friends (laughs). To eat, go outside, watch movies. I joke around with my brother, friends, and mom. I don’t like being sad.”* (I3).

Physical well-being involves maintaining a healthy body through activities like exercise, proper nutrition, and self-care:


*“R: What else is important to live well, to have a good life?*


*S1: Eat properly and according to our body’s needs, walk regularly and stay active. Because I have a stomach problem, I can’t eat and drink what I want and what I like.”* (I1).

These two aspects are inherently interconnected; emotional stability can enhance motivation for physical health, while physical fitness can boost self-esteem and emotional resilience. Together, they form *emotional–physical well-being*, the *Second Domain of O-QoL*:


*“R: How could you improve your life? What could you do to make your life better?*


*S2: I should stop listening to others and not pay attention to what they say that makes me feel sad and worthless. I should do things that I enjoy, focus more on myself and my body, because I like being fit; it makes me feel happy about myself! I love doing sports, and I really want to meet new people, travel, study, and also help others.”* (I2).

Emotional–physical well-being was associated with emotional health [47], physical activity [48], and physical health [49]. In our study, emotional health was associated with a sense of control over life [50], the ability to make one’s own decisions [50,51], the availability of choices [52], autonomy [53], independence [54], and privacy [55]:


*“Q: How could you improve your life?*


*K6: …I am always feeling down. I want to find a partner to build my life with and be happy. It’s not easy. I have met someone, but they tell me he’s not right for me. I don’t want people telling me what to do or who is or isn’t good for me. I don’t want to be lied to; it makes me angry. A friend of mine lies to me, but I don’t want that. I want to know the truth about who is a good person and who isn’t, without lies.”* (I6).

At the same time, physical activity was associated with self-esteem [56], body image [57], and self-image [58]:
*“S12: I want to lose weight, you know? I think I should hit the treadmill, ride a bike, or do whatever to shed some pounds. I’d really like to get in shape, feels nice to have a shredded body, and it’s also good for the girls you know.”* (I12);
while physical health was perceived as the absence of disease:


*“R: How could you improve your life, make it better?*


*S1: By having a good health. If only I didn’t have epilepsy and get easily disoriented or losing track of time. I could then easily live alone in the other apartment where my siblings used to stay.”* (I1).

*Material adequacy*, the *Third Domain of O-QoL*, refers to the sufficiency and quality of the material resources available to an individual in order to be able to participate in activities. It encompasses the physical environment and economic factors that contribute to a person’s overall QoL. Adequate material resources ensure that individuals have the necessary means to maintain privacy, comfort, and personal space, as well as the ability to meet basic needs and pursue life goals:

*“S1: It’s really difficult, I hate I don’t have my own room; I stay in the living room, I sleep there. I sleep in the living room. God help us call this a home (laughs). If you come over for coffee, we’ll sit where I sleep (laughs). I would like to have my own room, my personal space, so I can be alone and not always be with everyone else in the house.”* (I1).

Material adequacy was associated with socioeconomic factors, such as housing and living conditions [59]:
*“R: What do you do in your free time*
*S7: I lie down and have my coffee on the balcony; it’s nice at the group home. I live there with my sister; she’s now in her own separate apartment, and also there are four other people, so there are six of us in total.”* (I7);
as well as financial status [60]:
*“S1: How can you talk to anyone when you’re dressed like this? I feel terrible wearing this shirt; my sister-in-law and her brother bought it for me when I was 22, and I’m still wearing it. I can’t believe I still fit into it. If I dressed better, I’d be more confident and could chat with people on the street and make friends. If I had my own money, I’d buy clothes I like, … and overall, I think things would be a lot better.”* (I1);
and political context [61,62,63]:

*“S10: …I can’t go out anymore, I stay inside the house because I am afraid of the police! I can’t understand how to legally transport from one place to another, TV instructions are so confusing, so I can’t go out, I am afraid. I can’t even go to my friend’s house; she lives next to us, and I used to go there for coffee. I don’t see my friend anymore; she lives two blocks away and I can’t visit her. Will ever the coronavirus goes away? Neither I want it, nor my father, nor my mother. I want to ask about the coronavirus, if you know when it will end (laughs). Will we be wearing masks in the summer?”* (I10).

#### 3.2.2. Occupation: The Core Component of the Model of Occupational Quality of Life

Our research confirmed the overall positive influence of occupations on the lives of adults with IDs [64]. For this particular group of people, engagement and active participation in activities seem to contribute substantially to well-being, supporting the concept of occupation being the core component of QoL. The passage suggests that for adults with IDs, occupations are not merely tasks to fill time but are integral to their overall well-being:

*“S13: A good life is having occupations, that is. Having things to do.”* (I13).

Specifically, based on the perceptions of the respondents, participation in activities is a means of exposure to and familiarization with new life experiences [65] and pleasant passing of time [66]:
*“S10: In occupational therapy I prefer puzzles, but now we make ornaments and decorations for the Christmas tree, I do like making ornaments and decorations in general now. I pleasantly spend my time there with all the craftmaking and the conversations.”* (I10);
with multiple benefits for emotional–physical well-being [48]:
*S3: Some days, I’m happy when there’s work to do. But usually, I get bored and just chat with friends or play on my phone. I like helping out when there’s something to do; when there’s work, it’s good! But when there’s nothing to do and we just sit around, it’s boring. We just sit, sit, and sit. Time drags on, my mind aches, and it’s hard.* (I3);
as the individual assumes/performs roles [67] and creates a positive self-image [68]:
*“S7: Yesterday, we set up the social worker’s office—shelves, bookcases, the whole deal. I did it all by myself, no joke! I’m the best at this. I handle everything solo! I even took pictures! Got them right here on my phone, want to see? The teacher trusts me to do everything on my own. Yesterday, I was sanding down the shelves, all by myself! The social worker saw it and said it looks awesome. I can sand, glue, you name it! I’m really good at this stuff.”* (I7);
engages in entertaining activities and experiences positive emotions [66]:
*“S1: …The games we play on the TV are super fun, though they can be a bit challenging. I remember playing baseball on the Wii; it’s a bit tough with the bat, but we had a great time! I liked being split into teams, keeping score, and I was really pumped when we won. Of course, we can’t always win; sometimes we lose too (laughs)!”* (I1);
becomes motivated, and stays active [69]:


*“R: How do you evaluate your life?*


*S13: It’s good. The school helps a bit too, since I come here and do stuff. I like gym class the most, which is why I keep coming to school. I’d love it if we had gym class all day!”* (I13).

And, at the same time, facilitating interactions with other people supports social participation [70]:

*“S5: …I love going out, hanging out at the square, and chilling with my friends. Sometimes we play football at the church courtyard. My sister and I also hit up the Mall, do some shopping, and grab a bite at restaurants; just the two of us or with one of her friends. More than anything, I really enjoy going out and meeting new people.”* (I5).

Participants also referred to several occupations they considered important for their lives. Most respondents identified social leisure activities (e.g., going out with friends, going for a walk outside with friends, etc.) as particularly beneficial:

*“S9: …I have a few friends who live nearby, and sometimes we go out for a walk or we just hang out. But usually, I just go for walks around the house by myself. My mom doesn’t let me go farther than that; it’s not allowed. When I go out with my parents, we sometimes go a bit farther, and occasionally some friends come along too.”* (I9).

Significant reference was given to physical health routines (e.g., physical exercise, sports, healthy diet, and sleep), which also seem valuable in terms of respondents’ self-image:

*“S11: I love to exercise; I work out every day. I want to build muscles (laughing)! I want to look good so that I can be attractive to girls. I do track and field, running, ping-pong.”* (I11).

Furthermore, arts and crafts appear to be a means of emotional expression and discharge for female participants:

*“S6: I know how to draw well; I feel good when I draw. I usually draw little faces, hearts, people, and things like that. I keep my drawings in a folder, and one day I’ll bring them to show you what I’ve drawn. When I’m drawing, I feel my heart beating, and whatever I do with drawing gives me energy. It helps me to unload, I empty out all the things that weigh me down.”* (I6).

In addition, browsing the internet, watching movies, communicating on social media, and online gaming seem to support socializing and connecting with peers:

*“S12: I love playing video games with my friends. We have a blast and joke around a lot (laughs). It’s really fun! We can’t do as much in person; we don’t have the same freedom as we do in online games.”* (I12).

Finally, the VEF’s educational and vocational training programs were also positively valued. This passage reflects the positive impact of structured environments like school, particularly for individuals with IDs:

*“S9: …It’s nice to come here to school … I love being in the Carpentry workshop, coming to school every day and not staying at home.”* (I9).

However, participants tended to engage in more solitary activities (e.g., going out for lonely walks [4]), and rarely participated in social activities outside of the VEF (e.g., social interactions with relatives, friends, neighbors, or friends from or outside the VEF):

*“S6: I go out and take walks by myself. I want to see lots of people and make more friends. I’d love to go for coffee and shopping. But I’m alone, so I just stay home all the time. I have friends at school, but we don’t hang out outside of school, and I don’t really want to either.”* (I6).

They were involved in occupations mainly in the VEF, where they usually attended (i) therapeutic/educational programs: occupational therapy, speech therapy, psychotherapy, special education, etc.; (ii) vocational training: carpentry, bookbinding, cooking, etc.; (iii) physical education/sports; and (iv) arts and crafts: painting, music, theater, etc. And, at the same time, the VEF also seemed to be the main or only place for social interactions [45]:

*“S4: I like listening to music; it helps me take my mind off things. Also, I enjoy chatting with the teachers. I like everything we do here at school and all the activities, plus all my friends are here.”* (I4).

On the contrary, participation in desirable activities during leisure time was quite limited. In their free time, participants were mainly involved in solitary and passive activities [71] (e.g., watching TV programs, listening to music, playing with the mobile phone, etc.):


*“R: How do you spend your time at home? What do you do?*


*S13: I grab my phone and browse the internet. In the afternoons, I take a walk around the square. Then I head back home, have dinner, watch TV, and play games on my phone.”* (I13).

While, at the same time, their lives were dominated by inactivity and repetitive routines (e.g., eating [72] and occupational deprivation):


*“R: How do you spend your time at home?*


*S8: I eat the cologne, I eat the shaving foam (laughs), I watch TV, I don’t do anything. It’s better at school, here in the workshop with my teacher who loves me, she’s nice. At home, I’m constantly eating food; when my mother talks on the phone, I open the fridge, having olives, salami, cheese, I’ve devoured everything. I go to the playground sometimes, but I would like to go for a swim at the sea.”* (I8).

Moreover, participants were obliged to perform tasks, such as taking care of the household (e.g., washing dishes, general cleaning, cooking, etc.) and their parents/relatives, that occupied most of their free time and made it difficult for them to engage in desirable activities (e.g., physical exercise/sports):


*“R: Why can’t you go to the gym, exercise at home or listen to the music you like if you want to?*


*S6: Because there’s no time! I do all the chores. I manage the whole house; my dad cooks but I do everything else, I sweep, I clean.”* (I6).

Certain aspects of occupations positively supported QoL (Table 4). Specifically, social interaction and participation, along with the positive emotions resulting from satisfying occupational experiences, appeared to enhance QoL:


*“R: What are the things that affect your life, whether good or bad?*


*S12: I’ve got my crew, and I chat with the staff here. We chill with my mates, hit up the yard to play some basketball or football … I’m on the school football team, and I play defense. I don’t remember all the details though. I’ve gotten better at shooting now and have scored way more goals. It’s a blast; we play against other teams here and outside the school. Sometimes we lose, sometimes we win, but we always have such a great time, and that makes me really happy!”* (I12).

Additionally, factors such as a sense of autonomy, privacy, security, and control over one’s life, along with opportunities for free decision-making and availability of choices, were also identified as significant influences:

*“S1: …to be able to live in place where you will feel safe and confident, where no one else is there but only you. To be able to choose what you want to do, when to do it and with whom. To listen to music, relax and not having other people’s eyes on you all the time.”* (I1).

On the other hand, some other aspects of occupation had a negative impact on QoL (Table 4). Specifically, behaviors such as patronizing the individual, treating them as inferior, and criticizing or disapproving of their participation in activities, along with negative emotions arising from occupational deprivation and a lack of involvement in activities, adversely affected QoL:

*“S4: …I don’t want to leave home; I really want to stay with my family. But I wish we could talk more and have actual conversations. It’d be better that way. A lot of times, they don’t really get me, and we don’t sit down to talk. They don’t seem to care about what I like or don’t like, or what I need. I don’t know.”* (I4).

*“S10: …I feel crappy when I’m just sitting around at home, bored out of my mind. My mood’s not great. But I’m a lot better when I’m at school.”* (I10).

Finally, passive routines, inactivity, monotonous/repetitive activities, coercion, and imposition were also found as aggravating:

*“S1: …I don’t like that they make us do the same monotonous things all the time. Certainly, it is better than doing nothing, but I don’t want to paint and color all the time. I’m bored of it! But I’d rather do that instead of doing nothing.”* (I1).

### 3.3. The Influence of Socioenvironmental Factors on Quality of Life

The *environment/context* plays an important role regarding occupational engagement/participation, and therefore, its influence on participants’ QoL is crucial. In our study, three main environmental contexts were identified (Table 5):i.*Social environment* [3] (e.g., family relationships [44], social networks, and friends [43], as well as social beliefs and attitudes [73]);ii.*Economic/political environment* [74] (e.g., financial status, material adequacy, and financial independence, as well as health [61,63] and educational [62] policies);iii.*Personal environment* [75] (e.g., educational conditions in the VEF [76] and housing/living conditions [59]).

In our study, QoL was positively associated with the social environment in terms of personal relationships among family members and friends while engaging in joint leisure activities [44]:

*“S11: We play card games with my parents and my sister, like Taboo, and another board game I have at home called “Deal”. We play a bunch of different games. I also have this trivia game called “The Question Hunt”, which is super fun, and also Trivial Pursuit. I really like playing board games with my family and friends!”* (I11).

Furthermore, the economic environment [77] was also positively associated with QoL in terms of financial status and material adequacy for engaging in leisure activities:

*“S2: …We’ve had some financial problems, and it’s really affected our mood. When you have money, you can do everything; go to the gym, take swimming classes, learn new languages, travel, and basically do whatever you want.”* (I2).

Lastly, personal environment was positively associated with QoL via the educational environment [76] in terms of involvement in leisure, educational, and vocational training activities in the VEF:
*“S4: …I like a lot physical education, speech therapy, and occupational therapy. In speech therapy, we talk a lot about how I’m doing. I also enjoy talking with my psychologist; I like having discussions. In occupational therapy, I enjoyed playing on the Wii, especially the bowling game. I was pretty good at knocking down the pins. It’s fun to do different things besides just working at the Bookbinding workshop. Right now, in occupational therapy, we’re making Christmas decorations. They look really nice, and we’ll put them on the tree.”* (I4);
and via housing and living conditions [78] in terms of personal space and autonomy:

*“S12: I wanted to move to the island with my family, but I can’t because the space is really small, and there aren’t many things to do there, which isn’t great for me. I’d have to live with my grandma, and I can’t do that. I really want my own independence. If I had my own place and space, I’d be okay with moving back to the island. But right now, it just wouldn’t work for me.”* (I12).

However, it was also found that participants’ desired occupations (e.g., social interaction activities [79], employment [2], vocational training [80], academic advancement [81], and sexual expression [2]) are inhibited by a variety of socioenvironmental factors such as social stigma [73] and marginalization [2]:
*“R: …How do other people treat you about attending a special school?*
*S2: I never mention that I attend such a school, because they will assume that I am like retarded. But that’s not a case, I am not. I don’t know, anyone who hears about our school assumes it’s for people who have a problem. I hide it, I don’t talk about it, but of course two or three more guys are like me here.”* (I2);
dysfunctional/abusive family relationships [2]:
*“R: What would you like to have a good life?*
*S4: I’d like to be loved. I’m not sure what else I want. And I’d like to be able to do the things I enjoy.*
*R: What does it mean to be loved?*
*S4: It means that my mom and dad care about me. It’s important that they love us, that we get along well, and that they give us gifts. I don’t want to get hurt by my dad. Sometimes he gets really angry.”* (I4);
and the socioeconomic [82] and sociopolitical context [83]:
*“R: How are you doing?*
*S7: I’m okay, but not great. I am surviving. I don’t really have much. They cut off my allowance, so my brother-in-law gives me 20 euros now. It’s just a bit of pocket money, so I use it to buy a cheese pie or coffee. I get a little money, but it’s alright. What can you do?”* (I7);
including educational system policies [62,84] and social support services [85,86]:


*“R: Are there factors that limit you from improving your life or living it the way you want?*


*S12: Yes. Some teachers tell me that I should only do certain specific things, and none of these things I like. I want to do things that I want, like becoming whatever I want and taking care of my life myself, being responsible for myself. They tell me I need to become something good, but I don’t know what that is, I don’t know what they expect from me or what exactly I should do.”* (I12).

### 3.4. Quality of Life Assessment Criteria

Participants’ perceptions formed six categories of criteria regarding the assessment of QoL (Table 6). Specifically, the key criteria for assessing QoL were the following:

emotional state/health:
*“R: How do you currently evaluate your life?*
*S6: I’m doing okay. Sometimes when I come to school, I don’t feel like talking to anyone, but on other days I’m fine and greet everyone. Even though I get frustrated a lot, being here usually lifts my mood.”* (I6);
engagement in entertaining social activities:
*“R: How do assess your life?*
*S1: It’s kind of average. But if you come over to my place for coffee, it would definitely be better (laughs). It’d be great if someone would come over so I can hang out!”* (I1);
financial status:
*“R: How do you evaluate your own life?*
*S12: I have a good life, very good (sarcastically). No, it’s not (laughs); it feels like a mess. It’s terrible, it’s like trash (saddened). I’d like to get into an art school, but I don’t know what they’ll ask for, what supplies I’ll need to buy, or how I’ll manage the overall costs.”* (S12);
social relationships and social participation in the community:
*“R: What makes your life good and enjoyable?*
*S3: I love being happy and making jokes, and I enjoy hanging out and playing games with my friends at their places.”* (I3);
physical health:
*“R: How is your life?*
*S7: It’s great, I’m doing really well. I’m healthy!”* (I7);
housing, and living conditions:


*“R: …How is your life?*


*S1: Kind of crappy (laughs)? Kinda unhappy and miserable? Okay, I don’t sleep outside on the street, but I would like the place where I sleep to be proper home and not a hut! I want to have my personal space, and also heating because this isn’t a house, it’s the north pole!”* (I1).

### 3.5. Promoting and Improving Quality of Life

As a means of promoting and improving QoL, participants suggested the following (Table 7):

academic advancement [84] and employment [87]:
*“R: How could you make your life better?*
*S7: … I’d love to be a carpenter, but whatever job they give me I don’t have a problem. I told the social service that I will leave, and they replied that if I am a good student they will let me leave. That’s what they told me, they will find me work to gain my own money, not to get pocket money. To be able to get a coffee, a cheese pie, whatever I want!*
*R: How will this affect your life? If you find a job I mean.*
*S7: I want to find a proper job, not to be kicked out the next day, and I want to learn. I believe I’ll get a job; I believe so, I’m very good. I won’t sit at home all the time. I will get up every morning at seven o’clock and I will go to work, and then I will attend school in the afternoon. Or a carpenter or a waiter I will become, and I will go to school at night and get up in the morning for work! That’s what I want, something to fill my day.”* (I7);
engaging in social activities (e.g., going out with friends for a coffee [5]):
*“R: How could you improve your life?*
*S5: I can’t think of anything; nothing comes to mind. Going out with my friends would definitely help. I really like that, and I’ve missed it a lot!”* (I5);
sexual expression and satisfaction [88]:
*“S7: …Now I go to a brothel. It’s okay; I like it, it’s fine … I go there two or three times a month. However, I would prefer to have a girlfriend, a woman of my own. To meet her and marry her. I want to meet someone to be together with, to cook for me.”* (I7);
continuous participation in various activities (e.g., physical exercise/sports, therapeutic/educational programs, and video games):
*“R: How could you improve your life?*
*S6: I’ve never really thought about that. I want my life to be beautiful, but I’m not sure how. I’ve thought about wanting to do a lot of things. I’d like to do a lot more things outdoors. I want to learn how to dance. I like listening to music and being able to dance to it.”* (I6);
resilience skills [89], self-focus and personal needs satisfaction [90]:
*“R: How could you improve your life?*
*S11: Sometimes I look at myself in the mirror and say, “Oh God, help me!”, that’s the only way (laughing). I think about giving myself a shot in the ass, so I won’t have all these thoughts and won’t care about what people say.”* (I11);
social contribution and community service [91]:
*“S9: …I help the cleaners, put supplies in the storage room, and when the kitchen supplies arrive, I carry them and put them in the refrigerators. I like to help out at school. I would also like to help more often at the municipality’s soup kitchen.”* (I9);
strengthening family relationships [92] through participation in joint leisure activities:
*“R: How could your life be improved?*
*S7: …It would be nice if my father, my sister, and my brother-in-law could visit more easily. They can come, but we have to arrange it in advance. I’d like the process to be a bit simpler. They need to call before visiting, and it has to be approved by the housing facility. It’s a whole process. But my sister did come eventually! We sat on the balcony and had coffee together.”* (I7);
autonomy [93] in housing and daily living:


*“R: How could you make your life better? What would you like to do?*


*S1: …I’d love to have my own space, like being able to get up whenever I want, handle my own chores, and totally redo the apartment because right now it’s like a dumpster. I’d throw out all the old junk that’s piled up. I want to decorate my room my way, maybe tear down some walls, because they’re just crappy dirt and cement, and put in some insulation. Fixing up my room is super important, or even better, getting the other house sorted so I can live there. Also, I’d dump my brother’s old metal bed and get a wooden one.”* (I1).

## 4. Discussion

This qualitative study explored the perspectives on QoL of adults with IDs, utilizing the innovative theoretical framework of the MOJ and Critical Theory as the research instruments for the interpretation and content analysis of their narratives. A preliminary model of Occupational Quality of Life (O-QoL) was developed suggesting as the core component of QoL for personal/social development and self-improvement [94], the *occupation* [95], which comprises three interrelated environment/context interactive domains [11]: (i) social well-being; (ii) emotional–physical well-being; and (iii) material adequacy.

### 4.1. The Role of Occupational Engagement in Quality of Life: Centralizing Occupation

In the OT literature, there is substantial evidence suggesting that active participation in meaningful occupations significantly benefits QoL [3,28,96]. The occupations in which individuals engage shape their daily lives, influence their choices and goals, and thus play a crucial role in determining their overall QoL [28]. Meaningful engagement in these occupations is also associated with the fulfillment of essential psychological and intellectual needs, such as finding meaning in life [97]. A theoretical study by Causey-Upton proposed a model of QoL specifically for older adults living in long-term care facilities, with a focus on leisure activities as a central component of QoL [95]. While this theoretical approach provides important insights and is similarly guided by the principles of the MOJ, it remains a conceptual exploration rather than an empirical one. In contrast, our study, while also utilizing the MOJ framework, is grounded in data collected directly from participants, which facilitates a practical examination of how a range of occupational domains collectively influences QoL. Specifically, our research explores not only leisure activities, but also other crucial areas (e.g., education, vocational training, and activities of daily living, etc.) Additionally, a key distinguishing feature of our research is the active participation of individuals with IDs throughout the research process. This involvement was crucial in enhancing this study’s quality and credibility, enabling us to explore the subjective concept of QoL in an inclusive and emancipatory manner [98]. By engaging with the lived experiences of our participants, we were able to gain deeper, empirically based insights into the impact of occupational engagement on QoL, particularly in a population that may face significant occupational challenges. In summary, while Causey-Upton’s work offers valuable theoretical contributions, our study extends this understanding by providing a comprehensive, data-driven perspective on QoL that considers the full spectrum of occupational areas, enriched by the direct involvement of a traditionally underrepresented population.

Even though Schalock et al.’s foundational conceptual framework of QoL in the field of IDs [11] is built on a broad spectrum of domains that encompass various aspects of life, including emotional well-being, interpersonal relationships, material well-being, personal development, physical well-being, self-determination, social inclusion, and rights, our alternative model of O-QoL presents a different perspective. Our model emphasizes occupations as the primary means of achieving QoL and breaks down QoL into three main domains (social well-being; emotional–physical well-being; and material adequacy). While Schalock et al.’s framework spans a wide array of life domains, our alternative perspective centralizes the role of occupations, arguing that active participation in meaningful activities is fundamental to achieving QoL. In Schalock et al.’s model, occupations are part of personal development and physical well-being, but they are not the focal point. The model of O-QoL, however, places occupations at the heart of QoL, asserting that engagement in activities fosters overall well-being. Both models recognize the importance of social and environmental contexts, but they approach them differently. Schalock et al.’s framework includes social inclusion and rights, emphasizing social inclusion, support, and protection of human and legal rights. The model of O-QoL also values social inclusion but emphasizes how socioenvironmental factors such as family dynamics, economic conditions, and political contexts influence occupational rights/participation, and consequently QoL. In conclusion, the model of O-QoL focuses on the transformative power of meaningful occupations within their environmental context emphasizing the decisive role of engagement and active participation in activities of everyday life.

### 4.2. Empirical Insights into the Impact of Occupational Participation: Exploring the Occupational Quality of Life Domains

Participants’ personal views on QoL confirmed the multidimensional nature of the concept, aligning closely with Shalock’s et al. model of QoL in the field of IDs [11]. According to Schalock et al., QoL is a multidimensional construct influenced by individual characteristics and external variables. The key areas of QoL are consistent across all people, although their value and importance may vary for each individual [11]. In our study, the social well-being domain [11] was associated with the interpersonal social relationships [99] developed with family [44], friends [43] and the community [45,46]. Research involving 529 parents of children or adults with IDs confirmed and reinforced the importance of social relationships with family and friends as key factors contributing to QoL [100]. Additionally, in another study based on semi-structured interviews with six mothers of adolescents with IDs, friendship was found to be important for adolescents’ QoL. However, maintaining well-functioning and lasting friendships requires parents’ effort [43]. Furthermore, the feeling of belonging and contributing to the community appears to be an important factor for the participants’ QoL, as they seemed to perceive QoL in a more collective manner and they were receiving satisfaction from being active and valuable members of the community [46].

The participants’ emotional and physical well-being [11] were interpreted as interrelated and interdependent [101]. The relationship between them was conceived as a dynamic interaction of emotional health [102], physical exercise [48] and physical health [103]. Emotional well-being was associated with participants’ ability to control [50] and have independence [54] over their life. Van Leeuwen et al. also found that locus of control is associated with self-esteem, hope, meaning in life, positive emotions, and higher levels of QoL [104]. At the same time, physical well-being was associated in our study with self-esteem [56] and body image [57]. Similarly, in Nayir’s et al. research, individuals’ good body image came out as a positive QoL predictor [105]. 

Lastly, material adequacy [11] was associated with living conditions [59], financial status [60] and political context [61,62,63]. In a Greek study involving two groups of adults with IDs attending sheltered workshops (*N* = 31; aged 19–48), one group lived in a boarding house (*n* = 20), while the other lived with their parents (*n* = 11). The results indicate that both groups demonstrated low levels of independence compared to other QoL domains, while those living with their parents had lower social inclusion than those living in the boarding house [59]. This supports our study findings that participants living with their parents have passive/solitary daily routines and participate less in social life. A review of economic factors influencing QoL found a variety of impacts of poverty on health (e.g., limited health care access); productivity (e.g., limited leisure activities); physical environment (e.g., overcrowded homes); emotional well-being (e.g., increased stress); and family function (e.g., inconsistent parenting) [60], which come in line with our participants’ narratives about the financial difficulties and restrictions they face. At the same time, regarding the political context and disability policies, people with IDs in Greece significantly cope with a lack of specialized public health services and limited educational and employment opportunities [61]. The Greek economic crisis and the austerity measures’ effects on inclusive education policies have expanded educational inequalities, while institutional regulations employ inclusive education as a tool for further assimilation and discrimination of students with disabilities [62]. Specifically, participants were feeling ashamed of attending the VEF and they were practically marginalized within the general educational community. Finally, in accordance with our study findings, significant differences in QoL for people with IDs were observed during the COVID-19 pandemic [63]. Areas such as social interaction, social participation, friend relationships, and leisure activities were negatively impacted by the government’s health policies, affecting QoL and exacerbating the existing challenges within the underfunded public health system [63].

### 4.3. The Impact of Specific Occupations on Quality of Life

Specific occupations such as leisure activities, physical health routines (e.g., physical exercise), and arts and crafts appeared to be important for participants’ QoL. In addition, communicating on social media [106] and online gaming [107] seemed to enable social life participation, although safety concerns were raised. Moreover, educational and vocational training programs were also mentioned as significant for QoL. A study of Croatian citizens (*N* = 4000) found that engagement in leisure activities enhances subjective well-being, with the importance of specific activities varying by age and gender [64]. In another study, participants (*N* = 1399) completed the Pittsburgh Enjoyable Activities Test among other measures, revealing that engaging in enjoyable leisure activities is associated with better overall QoL [108]. People with IDs have mainly the same preferences and wish for leisure activities as their non-disabled peers, while both genders favor physical exercise/sports and cultural activities [71]. Adults with IDs were assessed using a QoL questionnaire and physical fitness tests, revealing that better physical fitness is associated with higher self-reported QoL [109]. Regarding arts and crafts, a Chinese study on expressive arts-based interventions revealed a positive effect on the emotional and behavioral well-being of female participants; however, the reverse effect was observed in males [110]. This confirms our study findings about differences in activity preferences as female and male participants enjoyed participating more in arts/crafts and sports, respectively. Adults with IDs are interested in social media, using YouTube for entertainment, Facebook to connect with family, and Instagram for social interaction with strangers [106], and our study participants behaved similarly. Additionally, a phenomenological study of three adolescents with Autism Spectrum Disorder engaging in online video games revealed a strong desire to socialize and communicate in virtual environments, although challenges in being misunderstood, issues with identifying friends, and lack of social rules awareness both online and offline were also highlighted [107]. Finally, research investigating the relationship between subjective QoL and the quality of Vocational Education services for students with IDs in Greek public special vocational schools found a significant emerging association between the two and low levels of self-determination among the students [76]. Our study findings confirm this interactional relationship between vocational training and QoL, whereas the low levels of participants’ self-determination corroborate our belief that VEFs still operate as places of social exclusion.

### 4.4. Occupational Rights, Environmental Influences, and Quality of Life Outcomes

On one hand, occupational rights such as social participation, positive emotions through occupational experiences, autonomy, privacy, security, sense of control over life, opportunities for free decision-making, and availability of choices were found to be beneficial for participants’ QoL. Self-determination empowers individuals with IDs to make their own choices, take control of their lives, and pursue their personal goals, leading to greater satisfaction, independence, and overall QoL [50]. A study of individuals with IDs (*N* = 141) in Italy found that self-determination and social skills were key predictors of higher levels of QoL, with greater autonomy and opportunities to make choices, particularly in residential settings. Our study confirms those findings, as participants who resided in Supported/Independent Living Accommodation Facilities or lived alone at their own house reported greater autonomy, participated in more outside activities, and positively assessed their QoL. On the other hand, occupational injustices such as occupational deprivation/lack of participation, inactivity, monotony, repetition, and passivity in occupations were found to be aggravating. Furthermore, patronizing the individual, treating them as inferior, continuously criticizing performance, disapproving participation, coercion, imposition, and negative emotions also had a negative influence on QoL.

In our research, the environment/context appears to be important for occupational engagement and participation. Therefore, three main environmental contexts influencing participants’ QoL were identified: (i) social environment; (ii) economic/political environment; and (iii) personal environment. A study that also examined factors affecting the QoL of individuals with IDs (*N* = 1264) found that environmental factors such as living conditions and activities of daily living significantly influenced QoL scores [75], aligning well with our research findings. Specifically, regarding the social environment, participants’ QoL was associated with joint leisure activities among family members and friends. A Greek study found that shared leisure time between parents and children with IDs significantly enhanced parent–child relationship quality, and therefore QoL [44]. The economic environment was positively associated with QoL as the financial capacity that enables participants’ access to leisure activities. Lastly, participants’ personal environment was positively associated with QoL via the activities taking place in their educational environment [76] and their living environment [78]. An Israeli study of adults with IDs (*N* = 85) found no significant differences in friendships or feelings of loneliness between two groups of people living in community residential settings and foster families, but those in foster families were more independent in leisure activities. This study revealed a partial association between friendships, leisure participation, and QoL, highlighting the importance of living conditions in social interaction and overall QoL [111]. In our study living conditions were also highlighted by the participants; however, the key aspects were personal space and autonomy. 

On the other side, our study revealed that social stigma [73], among other socioenvironmental factors, such as deprivation, abuse, and violence, a priori excludes individuals from desired occupations (e.g., employment and sexual expression/satisfaction), as it hinders their participation in activities of choice only because they are labeled as “kids”, “retarded”, or even “aggressive/dangerous”, which therefore negatively impacts QoL. On the contrary, people with disabilities face a higher risk of harassment, violence, and abuse, while individuals with IDs frequently endure various forms of abuse, including physical injury, sexual assault, and emotional trauma [2]. Children with IDs are at greater risk of family violence and abuse, especially if they also have physical disabilities or self-destructive behaviors. Family socio-demographic factors and social support significantly impact parenting quality as well as the associated risks of abuse and violence [112]. Moreover, stereotypical beliefs that people with IDs are childlike and asexual negatively impact their opportunities for intimate relationships and parenting, and also undermine their sexual health and safety [2]. Furthermore, stigma also affects the presence of people with IDs in the labor market. Negative employers’ attitudes pose significant challenges to people with IDs regarding competitive employment [113]. Few people with IDs are conventionally employed; instead, most of them are occupied in day centers and sheltered workshops [73].

### 4.5. Assessing and Enhancing Quality of Life

The self-assessment of participants’ QoL revealed six key criteria, including individuals’ emotional state/health; engagement in entertaining social activities; financial status; social relationships; social participation in the community; physical health; and housing/living conditions, that align well with the domains of the O-QoL model. Social well-being encompasses individuals’ social relationships and social participation in the community, highlighting the importance of interpersonal connections and active engagement in social activities. The emotional–physical well-being domain includes participants’ emotional state/health, as well as their physical health, underscoring the critical role of emotional health and physical activity in overall well-being. Finally, material adequacy is reflected in financial status and housing/living conditions, which are essential for ensuring a stable and supportive environment.

To promote and improve their QoL, participants emphasized the importance of occupation-centered practices, highlighting several key areas of focus. They advocated for academic advancement and employment opportunities, recognizing that education and work can significantly enhance personal growth and self-esteem. Additionally, participants expressed a strong desire to engage in social activities and a variety of leisure and recreational pursuits, understanding that these interactions foster connections and enrich their daily experiences. The development of resilience and self-focus skills emerged as crucial, enabling individuals to navigate challenges and cultivate a positive self-identity. Furthermore, the ability to express and satisfy their sexuality was identified as an essential aspect of personal fulfillment. Participants also underscored the need for social contribution and community service, indicating a desire to engage actively in their communities and give back to society. Family engagement was highlighted as vital for emotional support and connection. Finally, achieving autonomy in housing and daily living was deemed critical for fostering independence and enhancing overall QoL, allowing individuals to make choices that reflect their preferences and needs.

### 4.6. Strengths and Limitations

This research study presents several strengths that contribute to the understanding of this proposed model of O-QoL in the field of IDs. Firstly, by employing a qualitative research methodology, the research captures rich, nuanced insights into the lived occupational experiences of individuals with IDs, allowing for a deeper understanding of their perceptions regarding activities of daily living and their impact on QoL. This approach emphasizes the subjective nature of QoL, aligning well with the O-QoL framework’s focus on meaningful occupations. Additionally, the findings provide a promising theoretical foundation for the future development of specialized assessment tools and OT interventions that can be grounded in the participants’ articulated needs and desires.

However, as preliminary research, this study also has notable limitations that must be considered. The reliance on a small, convenience-based sample drawn from a single VEF limits the generalizability of the findings. This issue restricts the diversity of occupational experiences represented in the research, potentially overlooking the broader spectrum of challenges and opportunities faced by individuals in varying contexts. Furthermore, the constraints imposed by the COVID-19 pandemic hindered participants’ involvement in the analysis phase, which could have enriched the quality and credibility of the findings through collaborative reflection. Additionally, the absence of external researchers for cross-validation raises concerns about the reliability and trustworthiness of the conclusions drawn from the data. Finally, although this study relied on a single researcher for coding, our approach was dictated by the practical constraints of limited resources and personnel availability at the time. Despite this constraint, the methodology employed aimed to deliver a balanced and rigorous analysis, contributing meaningfully to the understanding of QoL in adults with IDs.

Therefore, future research should prioritize employing more diverse and representative sampling methods to enrich findings and improve generalizability. Incorporating mixed methods approaches, including both qualitative and quantitative data, could provide a more comprehensive understanding of this model of O-QoL. Additionally, exploring the inclusion of perspectives from caregivers and professionals could offer a more holistic view of QoL. To further address concerns about trustworthiness and credibility, future studies should consider involving multiple researchers in the coding process or implementing interrater reliability checks. This would ensure greater consistency and rigor in the analysis, thereby strengthening the overall quality of the research findings. Expanding research to other marginalized groups, such as refugees, individuals with mental disabilities, older adults, and people serving prison sentences, etc., could also provide valuable insights into the universal aspects of QoL and further validate this model.

## 5. Conclusions

The model of Occupational Quality of Life (O-QoL) (version 1) offers an innovative and valuable alternative framework for understanding and enhancing the QoL of individuals with IDs. Through participants’ insights, this research highlights the integral role of meaningful occupations in promoting well-being. The three identified domains, (i) social well-being; (ii) emotional–physical well-being; and (iii) material adequacy, which compose the core component, occupation, underscore the multifaceted nature of QoL and the interconnectedness of various life domains. While this study lays a promising foundation for the model of O-QoL, it also highlights significant limitations, notably in sample diversity and external validation. Future research should employ more diverse sampling methods and integrate mixed methods approaches to enrich findings and enhance generalizability. Additionally, incorporating perspectives from caregivers and professionals will provide a more comprehensive view. Expanding research to other marginalized groups will further validate the model and offer broader insights into QoL. Embracing these advancements will aid in developing effective, occupation-centered OT tools and practices, ultimately fostering greater autonomy, social inclusion, and improved QoL for individuals with IDs.

## Figures and Tables

**Figure 1 ijerph-21-01186-f001:**
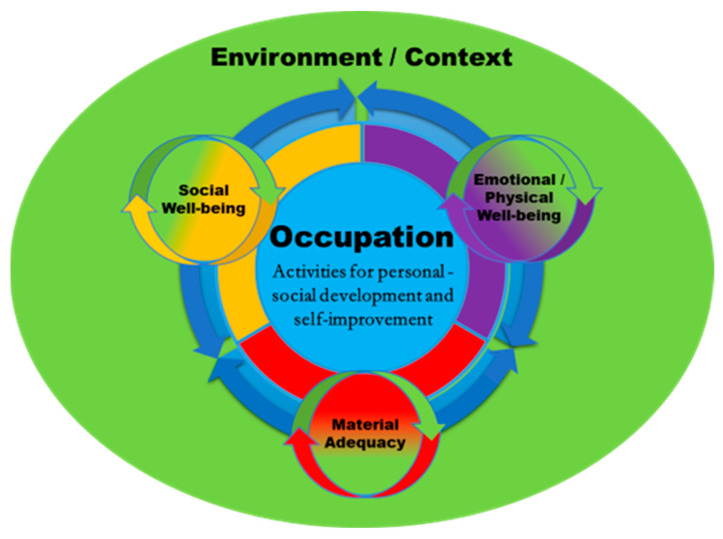
The model of Occupational Quality of Life (O-QoL) (version 1).

**Table 1 ijerph-21-01186-t001:** Outline of Interview Guide questions.

Topic	Example Questions
Key questions on occupational experiences and perceptions of occupational participation	In what kind of activities do you participate? In what kind of activities do you wish/not wish to participate? How do you participate in activities (what kind of roles do you take)? How do you feel when you participate/ not participate in activities?
Key questions on perspectives and perceptions of QoL	What does QoL mean to you? How do you assess your QoL? Which factors affect your QoL? How is your QoL affected by participating/not participating in activities?
Key questions on perceptions about ways to promote/improve QoL	How can your QoL be improved? How could you promote your QoL?

**Table 2 ijerph-21-01186-t002:** Demographic data.

Characteristics of the Participants in Frequencies (*N* = 13)	Interview Code	*n*	*%*
Gender			
Male	I2, I3, I4, I7, I8, I9, I11, I12, I13	9	69
Female	I1, I5, I6, I10	4	31
Age			
<21	I3, I9, I12	3	23
21–30	I2, I4, I5, I8, I11, I13	6	46
>30	I1, I6, I7, I10	4	31
Level of Education			
Primary Special Education School	I3, I4, I6, I8, I9, I10, I11	7	53
Special Vocational Education and Training Workshop	I2, I5, I7, I13	4	31
Primary School	I1	1	8
Lower Secondary School	I12	1	8
VEF’s ^1^ Training Workshops			
Carpentry	I3, I7, I9, I13	4	30
Bookbinding	I4, I5, I10, I12	4	30
Cookery	I6, I8	2	16
Candle Making	I11	1	8
Arts and Crafts	I1	1	8
Wedding and Baptism Supplies	I2	1	8
Total Years of Training in the VEF ^1^			
<10	I2, I3, I5, I9, I12, I13	6	47
>10	I1, I4, I6, I7, I8, I10, I11	7	53
Living Conditions			
Living with Their Parents	I1, I3, I4, I5, I6, I8, I9, I10, I11, I13	10	76
Living Independently without Any Aid from Siblings or Caregivers	I2	1	8
Supported Living Accommodation Facilities	I12	1	8
Independent Living Accommodation Facilities	I7	1	8
Personal Space			
Yes	I2, I7, I12	4	31
No	I1, I3, I4, I5, I6, I8, I9, I10, I11, I13	9	69

^1^ Vocational Education Foundation.

**Table 3 ijerph-21-01186-t003:** Conceptualization of the model of Occupational Quality of Life (O-QoL).

Model of Occupational Quality of Life (O-QoL) (Version 1)			Respondents (*N* = 13)
Core Component: Occupation Participating in Leisure Activities, Vocational Training, Employment, etc.,			I1, I2, I4, I5, I9, I10, I13
**for** **personal/social** **development** **and** **self-improvement**	First Domain: Social Well-Being	Friends and social relationships	I2, I6, I7, I10, I12 I3, I4, I7, I8 I9, I11, I12
		Family environment	
		Social contribution	
	Second Domain: Emotional and Physical Well-Being	Emotional and physical health	I1, I5 I7, I8, I9, I11
		Sense of control, choice, autonomy, security, and privacy in life	I1, I6, I11
	Third Domain: Material Adequacy	Financial status and political context	I5, I10, I12
		Living conditions and housing	I1

**Table 4 ijerph-21-01186-t004:** Aspects of occupation influencing quality of life.

**Occupational Aspects Enhancing Quality of Life (+)**	**Respondents** **(*N* = 13)**
Social interaction and participation	I1, I2, I3, I4, I5, I6, I7, I8, I11, I12
Positive experiences/positive emotions	I1, I3, I5, I6 I7, I8, I9, I11, I12
Autonomy, privacy, security, sense of control over life, opportunities for free decision-making, and availability of choices	I1, I4, I6, I11, I12
**Occupational Aspects Degrading Quality of Life (−)**	**Respondents** **(*N* = 13)**
Patronizing, treating as inferior, continuous criticism, and disapproval	I1, I2, I3, I4, I7, I8, I11, I12
Deprivation/lack of participation and negative emotions	I1, I2, I4, I6, I7, I10
Monotony, repetition, passive routines, and inactivity	I1, I4, I10
Coercion and imposition	I1, I12

**Table 5 ijerph-21-01186-t005:** Socioenvironmental factors influencing quality of life.

**Socioenvironmental Factors with** **Positive Influence on Quality of Life (+)**		**Respondents** **(*N* = 13)**
Social Environment	Enjoying social relationships and joint entertaining leisure activities with family and friends	I1, I2, I3, I4, I5, I6, I7, I8, I11, I12
Economic Environment	Financial status, material adequacy, and governmental policies	I1, I2, I4, I5, I7, I8, I11, I12, I13
Personal Environment		
Educational Environment	Training and participating in entertaining educational and leisure activities in the VEF ^1^	I1, I3, I4, I5, I6, I9, I11, I13
Living Conditions	Personal space and autonomy	I1, I5, I7 I8, I12
**Socioenvironmental Factors with** **Negative Influence on Quality of Life (−)**		**Respondents** **(*N* = 13)**
Educational Environment	Training in the VEF ^1^ provokes social stigma and marginalization	I2, I3, I6, I7
Family Environment	Dysfunctional family relationships and/or verbal/physical abuse/violence	I3, I4, I8, I12
Political Environment	Social marginalization, discrimination, and occupational deprivation	I5, I10

^1^ Vocational Education Foundation.

**Table 6 ijerph-21-01186-t006:** Assessment of quality of life.

Quality of Life Assessment Criteria	Respondents (*N* = 13)
Emotional state/health	I1, I2, I3, I5, I6, I9, I11, I12
Participation in entertaining social activities	I1, I2, I3, I5, I6, I9, I13
Financial status	I1, I2, I5, I12, I13
Social relationships (family, friends, community, etc.)	I2, I3, I10, I11, I12
Physical health	I2, I7, I13
Housing/living conditions	I1

**Table 7 ijerph-21-01186-t007:** Quality of life enhancement.

Means to Promote and Improve Quality of Life	Respondents (*N* = 13)
Academic advancement and growth/employment	I2, I3, I5, I6, I7, I8, I9, I11, I12
Social interaction and social activities	I1, I2, I3, I5, I7, I11, I12
Sexual expression and satisfaction	I2, I3, I6, I7, I11, I12
Continuous participation in various activities (e.g., physical exercise activities, sports, therapeutic/educational programs, and video games)	I1, I2, I3, I5, I6, I13
Disability elimination, resilience, and focus on personal needs	I1, I2, I3, I6, I4, I11
Social contribution/community serving	I2, I9
Family relationships/joint social activities	I4, I7
Housing, living conditions, and autonomy	I1, I7

## Data Availability

The data sets presented in this article are not readily available because this would compromise participant confidentiality.

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
