# Peer review of "Perspectives of Adults with Intellectual Disabilities on Quality of Life: A Qualitative Study"

_ijerph, 2024, doi:10.3390/ijerph21091186_

Round 1

Reviewer 1 Report

Comments and Suggestions for Authors

The article makes a valuable contribution to the field of quality of life for adults with intellectual disabilities, providing important insights from the individuals' own perspectives. However, there are key areas that can be improved to enhance the validity, representativeness, and transparency of the findings. 

Strengths

  1. The study addresses a crucial and under-researched area—the quality of life (QoL) of adults with intellectual disabilities (ID) from their own perspectives. This focus is essential for developing policies and practices that genuinely reflect the needs and desires of individuals with ID.

  2. The use of semi-structured interviews allows for in-depth insights into the participants' experiences and views, providing rich, detailed data that would not be possible through quantitative methods.

  3. Employing the Model of Occupational Justice provides a solid and well-established structure to the study, enhancing the credibility of the findings and facilitating the interpretation of the data within a relevant theoretical context.

  4. The interviews cover a wide range of topics related to occupational participation and quality of life, allowing for a holistic understanding of the participants' experiences.

Areas for improvement

  1. While convenience sampling is practical, it can introduce biases. Considering more diverse and representative sampling methods could enrich the findings and make them more generalizable.

  2. The description of some methodological aspects is somewhat brief. Providing more details on how the interview guide was adapted after the pilot and how coding discrepancies were resolved could strengthen the study's transparency.

  3. Although the use of NVivo for content analysis is mentioned, a more detailed description of the analysis process and the justification for the thematic categories chosen would be beneficial. Explaining how thematic categories were selected and refined could enhance the clarity and robustness of the analysis.

  4. The challenge of obtaining informed consent from individuals with ID is well-known. It would be useful to explain more clearly how participants' understanding of the study was ensured and how potential ethical conflicts were managed to guarantee the validity of the consent.

Suggestions for improvement

  1. Providing a more detailed description of each stage of the methodological process, from the preparation of the interview guide to data analysis, would include greater transparency and enable other researchers to replicate the study more easily.

  2. Including a more exhaustive and detailed analysis of the data, clearly explaining how thematic categories were derived and providing specific examples from the transcripts, would help readers better understand the study's processes and conclusions.

  3. Describing in more detail how ethical considerations were handled and what measures were taken to ensure participants' well-being and understanding would strengthen confidence in the study's findings.

  4. Proposing future research that uses mixed methods (qualitative and quantitative) to validate and complement the qualitative findings would be beneficial. Additionally, exploring the inclusion of perspectives from caregivers and professionals could provide a more comprehensive view of the topic.

Author Response

Dear Reviewer1,

Thank you for your valuable feedback on our manuscript, titled “Adults’ with intellectual disabilities perspectives on quality of life: a qualitative study”. We deeply appreciate your insightful comments and suggestions, which have greatly contributed to enhancing the quality and rigor of our study.

We have carefully addressed each of your comments and suggestions to improve the manuscript. To facilitate your review, we have highlighted the changes made in response to your feedback using a color-coded system in the revised manuscript. Specifically:

  • Blue highlights indicate changes made in response to your comments.
  • Green highlights reflect adjustments based on the feedback from Reviewer 2.
  • Yellow highlights denote areas where we addressed comments from both reviewers.

In summary, we have implemented the following revisions:

Sampling Methods: We have revised the manuscript to include a more detailed Limitations and Future Research regarding the sampling methods, in order to address the concerns about representativeness and biases. Our revised Discussion and Conclusion section now provides greater transparency on the steps we need to follow in the future to ensure a more diverse sample.

Methodological Details: We have expanded the description of the methodological aspects, including the adaptation of the interview guide post-pilot and the resolution of coding discrepancies. The revised manuscript now offers a clearer account of these processes.

Data Analysis: We have included a more detailed description of the content analysis process using NVivo, including how thematic categories were selected and refined. Specific examples from the transcripts are now provided to enhance clarity.

Ethical Considerations: The manuscript now includes a more detailed explanation of how informed consent was obtained and how ethical considerations were managed to ensure participants' understanding and well-being.

Future Research: We have incorporated suggestions for future research, including the use of mixed methods and the inclusion of perspectives from caregivers and professionals to provide a more comprehensive view.

We believe these revisions address your concerns and significantly strengthen our study. We are grateful for your constructive feedback and hope that the revised manuscript meets your expectations.

Thank you once again for your time and expertise.

Respectfully,

The Authors

Reviewer 2 Report

Comments and Suggestions for Authors

This is a very interesting paper, and approaches quality of life from a uniquely occupational perspective. It is a bit interesting in that it attempts to build an ambitious theoretical model based on interviews with a small, focused sample. It could work as a preliminary or provisional model, and is certainly unique in the field, but there are some important issues concerning the methodology and how this work is positioned that need to be dealt with.

A few points on the various sections of the paper…

Introduction

It is interesting that the authors don’t explore in the background much about quality of life (QoL) as a construct, therefore raising concerns about the depth of understanding of this complex and rich construct, and how, therefore, the emerging findings relate to it. I would suggest to build a case for the importance of the research, some of the elements included in the Discussion should be shifted to the Introduction. 

I’m not clear what the context is for this study. It is presented as a work environment for people with ID, and the educational and training background of each participant is well presented – yet it seems to be a treatment center as much as it is a workplace. (As a side-note, the authors in the introduction refer to ‘various environmental contexts’ of OT - yet provide two work-related contexts (vocational training, sheltered employment) rather than the broader context for the profession – which leaves the impression that OT is primarily about work – which it may be in this particular setting, but not in the profession in general). I also find the overall descriptions of occupational therapy and occupational justice/injustice to be a bit hard to follow for a lay (non-OT) audience.

It is incorrect to say (p. 2) that the international scientific research on QOL has not focused on the ID population, as there has actually been quite a bit of work in this area. It’s true that it has not been occupational focused, but as written, the statement is incorrect. As noted, the authors themselves cite some of the research in this area in the Discussion when they refer to Shalock’s foundational work on QoL as well as Boehm and Carter and others.  

Methods

Overall, the methods make sense, but there are several concerns:

-         Although the positioning of the work is a bit arbitrary (meaning the researchers seem to take a ‘critical theory’ type of theoretical stance that posits that persons with ID experience several affronts to occupational justice, namely occupational deprivation, occupational alienation, and occupational imbalance). It’s ok to take a specific stance of this type, but it should be clear that this is the theoretical approach, and/or there should be more than one or two references saying that persons with ID experience these types of exclusion (for example, other research on occupational justice has taken an affirmative approach, beginning with understanding the occupational lives of participants and what makes them feel happy, included, etc. rather than starting from a position of deprivation).

-     -          I assume the interviews were done in the Greek language, but there is no discussion about how the translation to English was done, and how equivalent terms were selected.  One wonders as to whether all the participants truly understood the scope of the questions about QoL. The authors mention ‘well being’ as an alternate term used – but that would not be the Greek term, and it, too, is an abstract term that would have little meaning for people in this population. I’m sure there were other Greek terms that were used to convey this concept, but the writing is very transparent about how this was achieved. In English we sometimes use terms like ‘a good life’, ‘a happy life’, ‘a fulfilling life’ – but the choice of term has to be clear. I note that one of the English translations from a transcript uses the term ‘a good life’.

-          The method described for assuring trustworthiness is not one I have ever seen in qualitative analysis (i.e. ‘the researcher independently coded the same transcripts twice. The compared coding showed 86% similarity, and any discrepancies were resolved through reflective thinking’. This sounds like a single researcher did the analysis, with no peer review or member checking, and given the very short description of the themes, raises some question as to the depth of the themes underlying what is a very well developed model. There is no reference provided for this approach to achieving trustworthiness, and I have not heard of it in many years of doing and teaching qualitative analysis. Overall, the method of content analysis is not really referenced or described beyond a single ‘how to’ article.

Findings

The reporting of findings is a bit sparse – such that there are few examples to speak to the trustworthiness of interpretation. The reporting jumps almost immediately to the model, and this can be a good way of introducing the audience to the overall findings, as long as there is sufficient depth in fleshing out the concepts and how they were arrived at. I appreciate the inclusion of Tables 3 - 7, which list all the themes – as well as, I think, which participants had comments that relate to the theme – and yet listing the number of people who said something about a particular theme doesn’t provide the depth of understanding we need, and is an unusual approach in a qualitative study. The reader needs more description of the theme itself, and different perspectives on it.

On page 9 there are a number of statements about certain elements of the model being ‘associated’ with QOL, the language being reminiscent of a quantitative study. The findings may be quite valid, but I don’t understand the numbers that follow the statements (I think perhaps they are linking to the literature sources, but it doesn’t really make sense to link almost every term in the findings to an outside source. This would typically happen in the Discussion section.  

Discussion

There is lots of good content here, and this is where the theoretical foundations of the study come through. If the Findings themselves can be enhanced to demonstrate that the model itself is well grounded in data, it would provide substance to the points raised. 

I find it a bit hard to interpret what is presented as the uniqueness of this research. This study is contrasted with another study that looked at QoL in an elderly population, with the current study being posited as different (better?) because it included consideration of all areas of occupation, and because of the population studied. I’m not clear what point is being made in this sentence: ‘the participation of people with ID in the research process enhanced the quality and validity of our study, as it made possible for us to explore this subjective concept in an inclusive and emancipatory way’. This perhaps suggests that previous research was not inclusive?  I suspect that it links back to the critical stance of the paper, such that the authors are saying that studying a (presumably) occupationally deprived population allows for greater depth of insight into the issue, but this line is not very clearly drawn (and difficult to substantiate). Again, this may be a problem of clarity of writing.

On p. 15, towards the end of the analysis, the occupation-based model is contrasted with Shalock’s model, saying, ‘Schalock et al.’s conceptual framework of QoL in the field of ID [80] is built on a broad spectrum of domains that encompass various aspects of life, including emotional well-being, interpersonal relations, material well-being, personal development, physical well-being, self-determination, social inclusion and rights. In contrast, O-QoL model emphasizes occupations as the primary means of achieving QoL’.  This is a fundamental switch in conceptualizing QoL, and as an occupational scientist, I find it appealing – but there has to be firm justification for why this is a plausible (perhaps superior) way of conceptualizing QoL and how to achieve it. That would be hard to do, but perhaps it is a case of more clearly stating that this is an alternative conceptualization that allows for a different approach to intervention (i.e. ensuring access to empowering occupations, etc.).  

Overall

The writing is hard to follow at times due to the awkward use of the English language, but if I step back and try to absorb overall the methodology and the analysis, the study seems to offer a potentially valuable model. The limitations are well articulated in the conclusion, but unfortunately undermine the credibility of the model presented here.

I think to properly assess the scientific quality of this paper on a resubmission I would want to see:

-          The English language writing revised to be more clear

-          A better introduction that fleshes out the existing conceptualizations of QoL that have been developed relative to an ID population (drawing information from the discussion up to the front of the paper)

-          More clarity on the methods, and a reputable approach to conducting content analysis (i.e. not sourced from a single article on content analysis from a single article based in emergency medicine)

-          Greater attention to the factors underscoring trustworthiness in qualitative research (basically any good textbook on the method will provide a guide to dependability, credibility BS confirmability) including rich description

Comments on the Quality of English Language

The English writing is quite good, but there are enough errors in word choice, grammar, and expression that it obscures some of the messaging. A good review by a native English speaker and some re-organization of the text would present the case made here better. 

Author Response

Dear Reviewer2,

We sincerely appreciate your detailed and thoughtful feedback on our manuscript, titled “Adults’ with intellectual disabilities perspectives on quality of life: a qualitative study”. Your insights have been critical in guiding us to refine and enhance our work. We have addressed all your comments in the revised manuscript and made substantial revisions to improve the clarity, rigor, and overall quality of our study.

To facilitate your review of the changes, we have used a color-coded system in the revised manuscript:

  • Green highlights indicate revisions made in response to your comments.
  • Blue highlights reflect adjustments based on Reviewer 2’s feedback.
  • Yellow highlights denote areas where we addressed comments from both reviewers.

Summary of Revisions

Introduction:

We have expanded the background on the concept of Quality of Life (QoL) to provide a deeper understanding of this complex construct and how our findings relate to it. Elements from the Discussion have been integrated into the Introduction to strengthen the case for the importance of our research.

Clarifications have been added in the Methods section to further explain the operation of the Vocational Educational Foundation (VEF) and the context of the study.

The statement about international research on QoL and ID has been revised to accurately reflect the existing body of work, highlighting the gap our study addresses.

Methods:

We have clarified the theoretical stance of the study and provided additional references to support the Critical Theory approach. We have also detailed the process of translating interviews from Greek to English and addressed concerns about term equivalency and participant understanding.

The methodology section has been revised to describe how trustworthiness was ensured in qualitative analysis, including more detailed procedures for coding and theme development. We have included references to established approaches for content analysis and trustworthiness.

Findings:

We have expanded the reporting of findings to provide more depth and detail on the thematic categories. Specific examples from the transcripts are now included to better illustrate the themes and enhance transparency.

The language has been adjusted to better reflect the standards of qualitative analysis. However, numerical references have been retained to facilitate cross-referencing with the discussion section, enhancing readability and minimizing potential confusion. If these references are deemed inappropriate, we are open to removing them.

Discussion:

The uniqueness of our research has been clarified, and the contrast with other studies, such as Causey-Upton’s theoretical study, has been better articulated. We have provided a more robust justification for our occupational-based model and its implications for QoL.

The writing has been revised to improve clarity and coherence, particularly in presenting the conceptualization of QoL and the potential benefits of our model.

Language Quality:

We have addressed most issues with language and grammar, ensuring that the revised manuscript is clear and well-organized.

We believe these revisions address your concerns and significantly improve the manuscript. We are grateful for your constructive feedback and hope that the revised version meets your expectations.

Thank you once again for your time and expertise.

Sincerely,

The Authors

Round 2

Reviewer 1 Report

Comments and Suggestions for Authors

 Dear authors,

The suggestions made have been largely incorporated. The article has improved considerably. Congratulations.

Author Response

Dear Reviewer 1,

Thank you for your congratulations and positive feedback on our revised manuscript, “Adults’ with Intellectual Disabilities’ Perspectives on Quality of Life: A Qualitative Study.” We are pleased to hear that you found the revisions beneficial.

We would like to inform you about the key changes made in response to the other reviewer’s comments, which have helped to further enhance our manuscript:

Sheltered Employment Programs: We updated the term from “sheltered employment programs” to “employment programs” to avoid over-emphasis on specific work settings and to reflect more current practices.

Terminology Adjustment: We corrected “further” to “more inclusive” on pages 2 and 3 to better convey the intended meaning.

Qualitative Methods: The reference to “interviews” as an example of qualitative methods was removed to streamline the text, as interviews are inherently part of qualitative research.

Critical Theory Paradigm: We revised the text to be more clear.

VEF Programs: The mention of “sheltered settings or in the open labor market” was removed from line 252, and the text has been clarified.

SPSS Analysis: We removed the reference to SPSS analysis, as demographic analysis is generally less relevant for qualitative research.

Tables and Quotes: Section 3.2 was revised to integrate additional quotes and better illustrate the themes.

Model Naming: The model title was updated to "Occupational Quality of Life (O-QoL) (version 1)" to reflect its preliminary nature and potential for future revisions.

Discussion Section: We streamlined the Discussion section by breaking it into sub-sections for clarity.

Language and Grammar: Minor punctuation and grammar errors were corrected, including adjustments to the use of semi-colons, commas, and articles.

These changes are highlighted in green in the updated manuscript. We believe they improve the overall quality of the manuscript. Thank you again for your support and encouragement.

Best regards,

The Authors

Reviewer 2 Report

Comments and Suggestions for Authors

-          Thanks for the many edits made in response to the feedback provided. The introduction section is improved and provides better background. There are a few minor changes needed to this section:

-          P. 2, line 64 – I would suggest removing ‘sheltered employment programs’ from the example – you would be better to just call it ‘employment programs’. This would remove the over-emphasis on work settings (and oddly, work in the community is not even mentioned). As well, sheltered employment is being phased out in many parts of the world, such that this point will ultimately date the article. I can understand that in Greece sheltered workshops may still be in full effect, and OTs may work in these settings, but that would not be the case in many countries.

-          P 2, line 93 I think you mean ‘more’ inclusive (rather than ‘further’). I see this error also on p. 3

Methods

-          P. 3, line 128 – It is not correct to say, ‘Qualitative studies such as interviews excel…’  Interviews are a means of data collection in qualitative studies. They are synonymous with the study method. You can just delete ‘such as interviews’.

-          P. 3, line 142 – I appreciate the fact that you re-framed your study as existing in the Critical Theory paradigm -but in the first sentence of this paragraph you mention two theoretical foundations that guided the design of the interview guide. Starting the next sentence with ‘It emphasizes’ does not work.

-          P. 5, line 252 – I don’t know that you need to mention ‘The skills acquired through VEF programs can lead to meaningful employment opportunities, either within sheltered settings or in the open labor market’ since this is already stated at line 232. You could edit the later sentence by just stating that employment may have a role in enhancing participants’ self-esteem and QoL if you wish to retain that.

-          P. 7, line 311 – I can’t imagine what demographic analysis conducted in SPSS for 13 participants would contribute to the interpretive analysis. Normally we don’t pool age, length of employment data or whatever to help interpret qualitative findings. If you figured  out a way to do this, you should explain it. Normally one would look at the array of demographics/attributes for each individual or across the group to help interpret whether certain participant ‘types’ (by age, gender, tenure in the organization, etc.) were more inclined to have certain views than others.  The only SPSS analysis I can imagine would fit might be cross tabs – but even still, the numbers would need to link back to individual cases. Personally, I would just take that out unless it was pivotal to the analysis in some way. No one expects to see SPSS come up in a qualitative paper.

-          P 7, lines 333 – 343 sound like findings to me.

-          The method of analysis and strategies used to ensure rigor of the findings is much improved. I see on p. 8 that you have referenced the intracoder reliability. It is quite an old reference (and approach), but if you did it you should leave this in.  I do take issue with use of the term ‘accurately’ (represented participants’ lived experiences).  Interpretations in qualitative research can never be ‘accurate’. Suggested alternatives would be ‘reasonably’ or ‘fairly’ – or just say that they were plausible representations of the participants’ experiences. And again, at line 400 – if your source uses the terms ‘validity and reliability’ you can use them, but most qualitative research uses terms such as ‘trustworthiness and credibility’.

-         Findings: It seems you have added much more text to introduce each individual quote in the first section, rather than building a deeper understanding of the themes by adding more quotes. It’s not necessarily to repeat the quote content in the text – the idea is to build more credibility for the theme by showing more examples. I think those long preambles could be removed. The addition of some new quotes is welcome, but rather than adding a single quote under each theme area (that sometimes doesn’t really fit) it would make sense to explain the theme in more depth, perhaps integrating some short phrases from different participants. For example, I have a hard time discerning the difference between ‘Personal / social development and improvement through activities’ and ‘social well-being’ in this section – and the quotes don’t help at all. I also don’t see how ‘Emotional well-being’ and ‘Physical well-being’ fit together into one category the way it is first presented.  Overall, the whole section 3.2 (Defining…) along with the Tables 3 and 4 showing how many people spoke to each theme seem unnecessary to me, as you flesh them all out later. In this section 3.2 they are just poorly defined, and yet you have a more fulsome description of each theme in the model starting on p. 12 (almost like you wanted to present quantitative results before getting to the qualitative section).  Why not just introduce the model at 3.2, showing the figure, and then use 3.2.1 and 3.2.2 to flesh out the themes as you have done?  You could move some of the quotes you added to section 3.2 to enhance the story you are telling in the descriptive sections. It would take a lot of the repetition out of this very long article, and not leave the reader confused as to what these themes are about in the first instance.

-          On Tables 3 – 7 in particular – I can see where these would have been useful appendices in a dissertation, and it was nice to see that you could substantiate where all the themes came from (that several people spoke to the themes and sub-themes) but without seeing the actual quotes and that they are relevant, they are not very useful, and such tables are rarely if ever seen in a published article. If you feel the tables are helpful in laying out the specific sub-domains of the themes, perhaps you could include just the theme areas in a smaller figure or combine them into one or two. But an example of where the depth is lacking is in your section titled ‘3.4. Quality of Life Assessment Criteria’ – where you are basically just providing a list of factors that were mentioned. If there isn’t enough information to write a full paragraph or two talking about how they determine what factors can be used to assess QOL (and how would they be different than the factors that define QOL?) then don’t include it. The final section on promoting and Improving QOL is equally weak.

-          P. 12 – Instead of calling the model, ‘The Preliminary Model of Occupational – Quality of Life (O-QoL)’ – which will ultimately need to change if you or other researchers do additional research on it, why not just call it ‘Model of Occupational Quality of Life (O-QoL) (version 1) or (beta version). As long as you mention in the text that this is a preliminary depiction of the model that may be subject to later change, it would be acceptable.

-      -          I took a cursory look at the other changes made to the Discussion and closing sections. They generally seem OK. The main advice might be to shorten the discussion section somewhat if possible, given the length of the manuscript. Part of this might be achieved by not repeating the findings so fully in places, and perhaps breaking up the Discussion into sub-sections so that it is clear what key points are being raised.

-          In summary, the main edits I suggest at this time are:

o   Reduction and refinement of the Results/Findings section as described above

o   Improvement of minor punctuation and wording errors.  

Comments on the Quality of English Language

The writing is generally very good. There just remain a number of small writing errors in punctuation and grammar to be addressed. The use of semi-colons is particularly problematic. For help, try reading this article: https://www.scribbr.com/language-rules/semicolons/  There are also examples of incorrect use of commas, missing articles, incorrect subject-verb agreement, etc. One particular punctuation edit is on p. 2, line 113 – where ‘ e.g. Schalock’s foundational research is significant contribution in the field ‘ should be in parentheses, not in the running text of the sentence. Another point is that when you pluralize ‘individual’s with ID ability’ it doesn’t work grammatically as such. It would need to be ‘the verbal communication ability of individuals with ID’ or the clumsier, ‘the individual with ID’s ability…’.  Also, the term ‘a research that examined’ as on p. 21 and perhaps elsewhere should read, ‘a study’ or simply ‘research’

Author Response

Dear Reviewer 2,

We greatly appreciate your comprehensive and insightful feedback on our revised manuscript, titled “Adults’ with intellectual disabilities perspectives on quality of life: a qualitative study”. We appreciate the time and effort you have invested in reviewing our work and providing constructive suggestions.

We have carefully addressed the changes you recommended. The revisions have been highlighted in green within the updated text to facilitate your review. Below, we provide a summary of how we have responded to each of your comments:

Sheltered Employment Programs: We have revised the text to refer to "employment programs" rather than “sheltered employment programs,” as suggested.

Use of “More” vs. “Further”: We corrected the terminology to “more inclusive” on page 2, as well as on page 3.

Qualitative Study Methods: We have clarified the reference to qualitative methods by removing the phrase “such as interviews” aligning with your suggestion.

Critical Theory Paradigm: We revised the text of the relevant paragraph.

VEF Programs: The mention of “sheltered settings or in the open labor market” was removed from line 252, and the phrasing was changed.

SPSS Analysis: We have removed the reference to SPSS analysis, as it was not pivotal to the qualitative findings. We agree that demographic analysis is less relevant in qualitative research, and we have refrained from including it.

Tables and Quotes: We have revised Section 3.2 and removed some long preambles. The tables have been retained, but we have ensured that they are now supported by relevant quotes to enhance their utility and relevance.

Model Naming: We have updated the model’s title to "Occupational Quality of Life (O-QoL) (version 1)" to reflect its preliminary nature and accommodate future updates.

Discussion Section: We have worked to refine the Discussion section, breaking it into sub-sections to clearly highlight key points.

Language and Grammar: We addressed the minor punctuation and grammar errors you identified, including correct usage of semi-colons and commas.

Regarding the intracoder reliability section, we retained it as we have conducted this procedure and believe it is important for transparency. Similarly, the tables were kept to substantiate the thematic analysis with supporting quotes.

We believe these revisions address your concerns and enhance the clarity and quality of the manuscript. Thank you once again for your invaluable feedback and for guiding us toward improving our work.

Best regards,

The Authors